# Mitochondrial Dynamics as Potential Modulators of Hormonal Therapy Effectiveness in Males

**DOI:** 10.3390/biology12040547

**Published:** 2023-04-03

**Authors:** Andrea Errico, Sara Vinco, Giulia Ambrosini, Elisa Dalla Pozza, Nunzio Marroncelli, Nicola Zampieri, Ilaria Dando

**Affiliations:** 1Department of Neurosciences, Biomedicine and Movement Sciences, Biochemistry Section, University of Verona, 37100 Verona, Italy; andrea.errico@univr.it (A.E.); sara.vinco_02@univr.it (S.V.); giulia.ambrosini@univr.it (G.A.); elisa.dallapozza@univr.it (E.D.P.); nunzio.marroncelli@studenti.univr.it (N.M.); 2Department of Engineering and Innovation Medicine, Paediatric Fertility Lab, Woman and Child Hospital, Division of Pediatric Surgery, University of Verona, 37100 Verona, Italy; nicola.zampieri@aovr.veneto.it

**Keywords:** andrology, hormonal therapy, mitochondrial dynamics, male infertility

## Abstract

**Simple Summary:**

Andrological diseases that affect patients in pediatric age represent important risk factors for alterations of their fertile potential in adulthood, and hence early diagnosis and treatment, even surgical and/or therapeutic treatments, are of primary importance. In this context, the discovery of the biological cues connected with alterations in the andrological sphere both in the pediatric age and adulthood could offer new insights into the identification of altered fertility potential markers or new therapeutic approaches. Mitochondria, the respiratory organelle of the cells, represent a key subcellular compartment within which the first enzymatic reaction of steroidogenesis takes place, thus highlighting that a correct arrangement of these organelles could be of crucial importance in both the correct hormone synthesis and response to hormonal therapies.

**Abstract:**

Worldwide the incidence of andrological diseases is rising every year and, together with it, also the interest in them is increasing due to their strict association with disorders of the reproductive system, including impairment of male fertility, alterations of male hormones production, and/or sexual function. Prevention and early diagnosis of andrological dysfunctions have long been neglected, with the consequent increase in the incidence and prevalence of diseases otherwise easy to prevent and treat if diagnosed early. In this review, we report the latest evidence of the effect of andrological alterations on fertility potential in both young and adult patients, with a focus on the link between gonadotropins’ mechanism of action and mitochondria. Indeed, mitochondria are highly dynamic cellular organelles that undergo rapid morphological adaptations, conditioning a multitude of aspects, including their size, shape, number, transport, cellular distribution, and, consequently, their function. Since the first step of steroidogenesis takes place in these organelles, we consider that mitochondria dynamics might have a possible role in a plethora of signaling cascades, including testosterone production. In addition, we also hypothesize a central role of mitochondria fission boost on the decreased response to the commonly administrated hormonal therapy used to treat urological disease in pediatric and adolescent patients as well as infertile adults.

## 1. Andrological Diseases Affecting the Fertile Potential

### 1.1. Childhood and Adolescence

There are different diseases of andrological interest in pediatric and adolescence age mostly associated with impaired fertility, including cryptorchidism and testicular torsion. Cryptorchidism is a condition in which one or both the testes fail to descend from the abdomen into the scrotal sac; it occurs bilaterally in one third of cases and unilaterally in two thirds of cases, with a frequency that is about 5% in full-term newborns and 30% in premature newborns [1]. Cryptorchid testes are classified on the basis of their position along the normal route of descent (high/low abdominal, inguinal, suprascrotal, high scrotal) or as ectopic when they are outside this route. In the clinical setting, however, a simple distinction between palpable and non-palpable and between unilateral and bilateral is most often used [2]. Interestingly, it has been reported that the expression abnormalities of genes that regulate testicular descent correlate with the risk of cryptorchidism, also including epigenetic aberrations at a post-conception phase [3]. In addition, brothers or sons of men with cryptorchidism are at a higher risk of developing it, with higher concordance rates among maternal half-brothers (about 6%) compared to paternal half-brothers (about 3.4%), suggesting an interest in potential future etiological work on maternal genes (particularly the X chromosome) and the intrauterine environment rather than paternal genes [4].

Although cryptorchidism is often considered a mild malformation, it represents the best-characterized risk factor for testicular cancer development and infertility in adulthood. Indeed, men with a history of cryptorchidism are frequently sub-fertile due to spermatogenic impairment, which is most frequently observed in bilateral forms [5]. Nowadays, the surgical approach, called orchiopexy, is the treatment of choice in 95% of the cases, with a percentage of success of about 100% [6].

Besides cryptorchidism, also testicular torsion represents an urgent condition that could take place during childhood/adolescence and may negatively influence fertility potential. It is characterized by the twisting of the testicle inside the scrotum with spermatic cord vessels getting occluded. Testicular torsion occurs in 1 out of 4000 males younger than 25 years [7] and the symptoms consist of acute scrotal pain and swelling, nausea, and vomiting [8]. Once the patient presents to the hospital, the physicians evaluate his condition and perform a physical examination; when torsion is suggested, immediate surgical exploration is indicated and should not be postponed, as there is a favorable range of four- to eight-hours for surgery intervention before the occurrence of permanent ischemic damage. Delayed treatment may be associated with decreased fertility or may necessitate orchiectomy [8]. Nowadays, besides surgical intervention, no therapeutic approaches are indicated during the recovery phase. However, very recently, we reported experimental results about the possibility to use human gonadotropin hormone (hCG), which is commonly prescribed to improve fertility potential in infertile men and to increase testicular trophism in post-operative cryptorchid patients, as adjuvant therapy also for patients with testicular torsion to improve testicle health [9].

### 1.2. Adulthood

Many of the diseases that occur in adulthood originate before the age of 18 and, sometimes, even during gestation [10]. Notably, risk factors that alter the fertility potential in males can be diagnosed and treated in pediatric/adolescent age, opening the way to the importance of prevention and preservation of fertile potential in young patients. Infertility is a clinical and social problem that affects an increasing number of people each year worldwide [11]: indeed, available data suggest that between 48 million couples and 186 million individuals have infertility globally. Infertility is a disease of the male or female reproductive system defined by the failure to achieve a pregnancy after 12 months or more of regular unprotected sexual intercourse. Primary infertility is the inability to have any pregnancy, while secondary infertility is the inability to have a pregnancy after a previously successful conception. It is estimated that up to 7% of men are affected by infertility and 50% of fertility problems within a heterosexual couple are due to the man [12]. It has been reported that over 30% of male infertility is due to testicular diseases, while 15–30% is related to endocrine defects, indicating that there are several conditions [12]. In other cases, the cause of male infertility is unexplained. Anyhow, male hypogonadism is commonly found in infertile men [13] and it is a condition characterized by insufficient testosterone (TT) production by Leydig cells, which are the testicular cells responsible to produce TT. According to the level along the hypothalamic-pituitary-gonadal (HPG) axis where the alteration occurs, hypogonadism can be central (secondary or hypogonadotropic) if the problem originates at the level of the pituitary or hypothalamus or primitive (testicular or hypergonadotropic) if the alteration happens on the testes. Furthermore, hypogonadism can be congenital or acquired and can affect all the testicular compartments (Leydig, Sertoli, and germ cells) or just some of them. Lastly, based on the period of life within which hypogonadism appears, it can be considered fetal, pre-puberal, or puberal [14].

Another disease that affects about 40% of infertile men is varicocele, which is the dilation of the veins of the pampiniform plexus, resulting in spermatic vein reflux, with blood stasis, increased temperature, and poor oxygenation. Varicocele generally affects the left side (78–93% of cases) and it is often asymptomatic, and therefore it is diagnosed by chance during a routine visit or by the patient himself, suspicious of the presence of a mass felt by self-examination of the testicles [15]. Three grades of varicocele are used in the clinic, and the grading is based on the ability to visualize and/or palpate the varicocele in both the relaxed states (grade II and III) or while inducing the Valsalva maneuver (grade I) [15]. During adulthood, the prevalence of varicocele can be about 45% among men seeking care for primary infertility, i.e., when a pregnancy has never been achieved, and 80% among men seeking care for secondary infertility, i.e., when at least one prior pregnancy has been achieved [15]. In certain patients, varicoceles can cause testicular damage, resulting in loss of testicular volume, spermatogenic dysfunction, disruption of hormone production, and sperm DNA damage [16]. In addition, other evidence reported an impairment of testicular Leydig cell function, resulting in decreased TT production; indeed, different studies have demonstrated significant TT level improvements in patients with hypogonadism after a clinical repair of varicocele [15]. Varicocelectomy is the only surgical procedure specifically designed to improve spermatogenesis. After this procedure, a significant improvement in semen analysis has been demonstrated in 60–80% of men and in pregnancy rates in 20–60% of the couples [17], due to an improvement in sperm density, motility, and morphology [18].

## 2. Hormonal Therapy for Andrological Diseases

Most andrological diseases affecting male fertility are closely linked to the function of complex organs, such as the pituitary gland and gonads, that produce and secrete hormones. Since hormonal supply is essential for the normal development of masculinity and subsequent fertility, currently preclinical and clinical efforts to treat andrological diseases are mainly directed towards hormonal therapies. Among these, testosterone replacement therapy (TRT), which has been developed in the last 70 years, represents an option to treat pathological conditions, such as testosterone deficiency and hypogonadism. However, there are conflicting data on the use of TRT, whose aim is to restore TT levels in the physiological range. Indeed, despite replacement therapy with TT generally showing significant short-term benefits in men with symptoms of low testosterone, TRT can exert side effects in long-term treatments, with risks and benefits that still remain unknown [14]. Some evidence regards the suppression of the HPG axis with an increased risk of infertility, together with a negative feedback action on follicle-stimulating hormone (FSH)-mediated stimulation of Sertoli cells [19], by further altering the endogenous TT production [20].

Therefore, other therapies, described below, show a higher effectiveness for treating andrological diseases, together with the improvement of fertility in males and the consequent amelioration of patients’ quality of life.

### 2.1. hCG and FSH

One of the most recently studied hormones in the context of andrological pathologies is hCG, a glycoprotein hormone belonging to the gonadotropin family and produced mainly by differentiated syncytiotrophoblasts [21]. It is involved in embryonic signaling and is a key element of gestations development during pregnancy [22]. The molecular weight of hCG ranges from 36 up to even 41 kDa according to the glycosylation of its various forms. It is composed of two subunits, α and β, linked by a non-covalent bond. The α subunit is common to the diverse forms of hCG and to the other gonadotropins, such as luteinizing hormone (LH) and FSH, but also thyroid-stimulating hormone (TSH) secreted by thyrotropic cells. Concerning the β subunit of hCG, it is structurally similar to the β subunit of LH. However, it has an extra 24-amino acid sequence called the C-terminal peptide (CTP) [23]. Due to their homogeneity, both hCG and LH activate multiple pathways via binding to the same luteinizing hormone/chorionic gonadotropin receptor (LHCGR). HCG mimics the effects of LH but about 10 times more intensely in adenosine 3′,5′-cyclic monophosphate (cAMP) activation and slightly more potently on cAMP-dependent phosphorylation of extracellular signal-regulated kinase 1/2 (ERK1/2) [24]. Serum levels of hCG change during the fetal life, chiefly the presence of hCG is detectable only throughout the gestation period, with a peak between the second and the third month (Table 1) [25]. Afterwards, hCG level decreases until birth and becomes undetectable for the rest of the male’s life [26]. The absence of hCG in the serum of children and adults renders its detection a valuable marker not only for pregnancy but also for some neoplasia, including testicular germ cell tumors, that produce and secrete this hormone [27], presumably due to the epigenetic regulation of its expression in cancer cells.

In addition to hCG, the usage of another gonadotropin, i.e., FSH, is widely studied. FSH is a heterodimeric glycoprotein released by the anterior pituitary and it targets gonadal cells in both males and females. The FSH-specific β-subunit is non-covalently associated with the common α-subunit which, as mentioned above, is also shared with the other members of the gonadotropin family [28]. In males, FSH released by the pituitary gland binds to the G-protein-coupled FSH receptor (called FSHR) expressed by Sertoli cells. In these cells, FSH stimulates spermatogenesis through the activation of mitosis and trophic effects, despite the fact that the complex network of signaling pathways is not fully known yet. Nevertheless, the deficiency of FSH signaling impairs fertility capacity, thus requiring clinical treatments to restore gonadal function. For this purpose, several FSH analogs were developed to modulate the target cell response to the hormone, especially for infertility treatments [29], as discussed below.

### 2.2. Hormonal Therapy for Cryptorchid Patients

Since the 1930s, in Europe, the most supported and used therapeutic approach for testicular descent failure was the administration of hormonal therapy, given the belief that gonadotropin and androgen deficiencies were one of the etiological factors at the basis of undescended testicle causes [30]. Despite the fact that the most frequently used hormonal therapy was hCG because of its capacity to mimic the effects of LH in Leydig cells, this method has been mostly abandoned because of a success rate percentage of less than 20% [31]. Indeed, without surgical correction, an undescended testicle would likely descend during the first three months of life; if it remains undescended, to reduce the risk of increased testicular germ cell tumors outcome, testicular torsion, inguinal hernias and to minimize the risk of infertility (especially bilateral cases), the testicles should be brought into the scrotum with surgery, i.e., orchiopexy [32]. This surgical procedure takes place between 6 and 18 months of age, since within this time window there is a lower risk of future complications. Nevertheless, the hCG role in support of testicle wellbeing is widely described and, thus, despite the fact that it is not even used as a first-line treatment anymore, it could be administered after surgery. Indeed, hCG adjuvant therapy has been shown to improve testicular vascularization, volume, and morphology safely [33]. Therefore, early corrective surgery is considered to be necessary, recommended, and based on an observational and follow-up study that analyzed various metrics in adulthood that have yielded favorable results regarding early orchiopexy and reproductive function [34].

### 2.3. Hormonal Therapy for Infertile Men

Male infertility is a frequent medical condition and is increasingly observed as a reliable sensor of future male health conditions, in association with cardiovascular disease, testicular cancer, quality of life, and increased all-cause mortality. In about 50% of the cases, infertility is idiopathic, which means that is has no known cause, and the remaining percentage is related to pituitary and gonadal alterations, including the aforementioned cryptorchidism, testicular atrophy, varicocele, and post-testicular causes due to secondary consequential alterations occurring in the testis [12]. The challenge for reproductive modern medicine is represented by early diagnosis and proper management of infertile men.

To evaluate testicular functionality, it is necessary to reck principally serum gonadotropins and TT levels together with spermatogenesis, functions that are mainly regulated by the HPG axis. Therefore, in most cases, patients should receive therapies aimed at improving the function of this complex hormonal axis, rather than treating their downstream problems. Some of the most used therapies for infertile men comprise selective estrogen receptor modulators (SERM), aromatase inhibitors (which are both involved in accurate regulation of intratesticular testosterone/estrogen ratio), hCG, recombinant follicle-stimulating hormone (rFSH), or a combination of hCG + rFSH, according to physicians’ first choices [35]. HCG is generally administered to patients with infertility associated with hypogonadotropic hypogonadism. It acts as an LH analog by binding to the receptor LHCGR on Leydig cells, stimulating them to produce and release intratesticular TT. In addition, other studies have also observed a decrease in triglyceride levels and an increase in lean body mass and muscle tissue in patients with hypogonadism after treatment with hCG [36]. On the other side, hCG monotherapy is not able to raise FSH levels since it has no documented effects through the binding to FSHR; thus, the use of combination therapies consisting of hCG + rFSH induces an increase of native FSH levels [37]. FSH promotes the production of androgen binding protein (ABP) in Sertoli cells and is crucial for maintaining spermatogenesis by keeping monitored concentrations of androgens in the seminiferous tubules [38]. Indeed, clinical studies on idiopathic infertile men reported an increase in sperm production in dependency on the FSH dosage [39]. The use of LH instead of hCG, in association or not with FSH, is still to be explored. Since LH and hCG are different hormones and are not fully swappable in their signal transduction properties, the use of LH in the treatment of male idiopathic infertility should be tested and could provide interesting results. Finally, it has also been reported that leukocytes can interlace with sperm, reducing sperm motility, and the chances of fertilization, thus making the evaluation of the impact of immune cells on fertility intriguing [40]. The aim behind all these efforts is to find a personalized therapy for all forms of male infertility with the least possible invasiveness.

### 2.4. Applications of Hormonal Therapy for Patients with Other Andrological Diseases

Up to now, as described above, the two andrological pathologies that are treated with hCG and/or FSH are cryptorchidism and infertility. The latter comprises endocrinopathies, including hypogonadotropic hypogonadism, and non-obstructive azoospermia [41,42]. However, how these therapies impact other andrological pathologies is still being defined. Recently, an in vitro culture study of a tissue closely linked to the testis, i.e., gubernaculum, showed promising data on the adjuvant hCG monotherapy for testicular and spermatic cord torsion for which no therapeutic approach is currently considered in the post-operative phase [9]. In this study, it has been shown that the in vitro growth of gubernacular cells derived from three different patients with diverse degrees of testis torsion depended on the severity of the damage; indeed, cells derived from a patient whose testicle was removed because too damaged were not able to grow in culture. In addition, gubernaculum testis expresses LHCGR, and thus it is sensible to undergo hCG therapy, showing an increase in cell proliferation. The advantage of the use of hCG also for this type of pathology is important, since clinical data show that it induces vascularization [33], thus further corroborating the advantages of applying this therapy in the case of testis torsion, in which due to rotation of the testis there is a consequent restriction of blood supply. In addition, another study showed that the treatment of rats with insulin-like growth factor-1 (IGF-1) and growth hormone (GH) improved germ cell histology, spermatogenesis, and fertility after experimental testicular torsion and subsequent detorsion after 6 h [43].

Thus, the application of already existing hormonal therapies or new ones could be considered in the clinical practice, prior validation in in vitro and in vivo models, in order to support a prompt recovery of the testicle in case of damages or pathologies, always keeping in mind that it would help the patient to avoid fertility problems.

## 3. Luteinizing Hormone Receptor

Proper functionality of the luteinizing hormone receptor is crucial in the case of hormonal therapy with hCG, mostly because it activates specific signal transduction pathways that support the function of Leydig cells. In this chapter, we deepen the signal transduction pathways activated by LHCGR ligands (LH and hCG) and the most common variants of LHCGR that could affect its functionality and, thus, could have a reflection of the fertility potential.

### 3.1. LH as Ligand

LH is a glycoprotein hormone that belongs to the family of gonadotropins, which includes FSH and hCG. Gonadotropins are coded by similar genes, and, thus, they share similar properties, including the expression of alpha and beta subunits. As described above, the alpha subunit is common between FSH, LH, and hCG, whereas the beta subunit is different and gives each hormone a kind of biological specificity [44]. Particularly, the alpha subunit of LH is made up of 92 amino acids and the beta is made up of 120 amino acids, and thus the combination of these two subunits has a total mass of 28 kDa [45]. LH is co-secreted along with FSH by the anterior pituitary. LH release is stimulated by gonadotropin-releasing hormone (GnRH) and inhibited by estrogen in females and testosterone in males. LH has various functions, which differ between women and men. However, in both sexes, it contributes to the maturation of primordial germ cells. In men, LH stimulates the Leydig cells of the testicles to produce testosterone, while in women it triggers the production of steroid hormones from the ovaries [46]. Once LH moves into the testicle, it binds to LHCGR, which is expressed by Leydig cells. LHCGR belongs to the G-protein-coupled receptor family and once triggered by LH, it activates adenylyl cyclase, an enzyme that increases intracellular cAMP concentration, which then activates a kinase molecule called protein kinase A (PKA). PKA is the main kinase responsible for the phosphorylation of specific transcription factors, including steroidogenic factor 1 (SF-1), cAMP response element-binding protein (CREB), CRE modulator (CREM), CCAAT/enhancer-binding proteins (C/EBPs), and activator protein 1 (AP-1), that are implicated in regulating the expression of the genes encoding for steroidogenic acute regulatory (StAR) proteins and steroidogenic enzymes [47]. In the end, the generation of cAMP by LH results in the synthesis of TT. Indeed, the cAMP promotes the transfer of cholesterol to the inner mitochondrial membrane where it is metabolized into pregnenolone via the P450 cholesterol side-chain cleavage enzyme (P450scc or CYP11A1) and, further, is converted to progesterone by 3β-hydroxysteroid dehydrogenase (3β-HSD). In Leydig cells, the maturation of progesterone to androstenedione is catalyzed by 17-hydroxylase/C17-20-lyase (CYP17A1). Finally, androstenedione is converted into testosterone by type 3 17β-hydroxysteroid dehydrogenase (17β-HSD3) [48]. In the serum, LH levels change along the male’s life (Table 2); indeed, during the first postnatal hours, the circulating levels of LH are very low. Subsequently, they increase during the first weeks of life until 3 to 6 months of life, which is typically known as “mini-puberty”. Afterwards, serum LH drops to non-detectable levels, remaining stable until puberal onset. Through puberty, from the ages of 9 to 14, LH levels increase and stimulate the proliferation of Leydig cells and the production of androgens even during adulthood [49]. As reported in Table 2, the normal range for men over the age of 18 is approximately 1.8–8.6 IU/L, at 50 years, it is 2.1/10.4 IU/L, and over 70 years, it is 2.2/11.2 IU/L [50].

### 3.2. hCG as Ligand

As already mentioned above, hCG binds to LHCGR as the “sister” gonadotropin LH. Indeed, the two gonadotropins are encoded by a common gene cluster and their protein structures are more than 80% homologous [21]. Aside from the aminoacidic sequences, carbohydrates bound to both hormones are crucial for their biological functions and the hepatic clearance of them from the body. In a recent study, it has been demonstrated that the removal of carbohydrates from hCG generates its de-glycosylated forms, which bind LHCGRs with the same high affinity as the parental molecule, but with partial or complete impairment of function. Interestingly, these de-glycosylated “antagonist” variants of the hormone have been detected in the sera of patients with chronic renal failure disease associated with hypogonadism [55].

The effects of LHCGR activation due to hormone binding are sex-specific and depend on the intracellular enzymatic apparatus of the target cells and, apparently, on the receptor expression levels. Although both two gonadotropins share multiple features, the hCG effect is different from that of LH based on their different interactions with the receptor at the extracellular hinge region; the diverse binding with the same receptor causes the LH and hCG activation of different signaling pathways [56]. In the male gonads, the activation of LHCGR by both LH and hCG conclusively results in the production of TT, thanks to the activation of the cAMP/PKA pathway described above. However, in the last decade, several in vitro studies demonstrated that hCG activates the steroidogenic cAMP/PKA pathway to a greater extent than human LH, which, instead, preferentially stimulates the proliferative and anti-apoptotic pathways of extracellular signal-regulated kinases and protein kinase B. Interestingly, the steroidogenic efficiency of this intracellular pathway seems to decrease with age, as an assumed effect of cumulative exposure to oxidative stress occurring during the individual’s life [57]. Furthermore, hCG also induces a more rapid and effective recruitment of the β-arrestin 2 compared to LH, suggesting that hCG exerts more efficiently the receptor desensitization [58]. Despite the remarkable progress in recent years, further studies on the molecular mechanisms triggered by LHCGR activation are needed; moreover, the intrinsic pro-apoptotic potential of these hormones, the existence of receptors assembled as heteromeres, and their expression in extragonadal tissues need to be investigated.

### 3.3. LHCGR Variants

LHCGR is a member of the class A guanine nucleotide-binding protein-coupled receptors (GPCRs) that, together with FSHR and thyroid-stimulating hormone receptors (TSHR), belong to the subfamily of glycoprotein hormone receptors (GPHRs). The overall organization of the *LHCGR* gene is highly conserved between species, consisting of 11 exons and 10 introns. In humans, as well as in primates, this conserved genomic organization is complemented by an additional primate-specific exon, the exon 6A, which lies in intron 6, thus resulting in a gene composed of 12 exons and 11 introns [59] (Figure 1). In addition, over the past few years, another splice variant has been reported in the common marmoset monkey (*Callithrix jacchus*), in which the wild-type form of LHCGR in this species completely lacks exon 10 [60].

Human LHCGR (*hLHCGR*) is located on the short arm of chromosome 2 (2p21), where exons 1–10 contain the code for the transcription of the signal peptide, N-terminal Cys-rich region, the leucine-rich repeat domain (LRRD), and the N-terminus of the hinge region that connects the LRRD to the serpentine-like transmembrane domain. The C-terminal segment of the hinge region, the seven transmembrane helices, the connecting loops, and the C-terminal intracellular part are encoded by exon 11 [61]. The full-length hLHCGR (named also type I variant), which contains exons from 1 to 11 including exon 6 (and not exon 6A), is composed of 699 amino acids, with a total mass of 85–95 kDa when mature and fully glycosylated [61]. Functionally, the full-length hLHCGR exploits its central role in a physiological aspect of gonadal maturation by binding both hCG and LH mainly on the cell membrane of the ovarian theca, granulosa, luteal, and Leydig cells [61]. Furthermore, there is evidence that cells that compose the gubernaculum testis may express LHCGR [9]. To achieve proliferation and differentiation of a variety of cell types, the primary pathways exploited by the activated LHCGR, once it couples with Gα_s_ G-proteins, are signaling pathways of cAMP/PKA, ERK, and protein kinase B (PKB/AKT) [56,61]. Furthermore, the presence of a high concentration of both receptors and ligands can lead to an increased intracellular Ca^2+^ through the activation of phospholipase C and inositol phosphate signaling [62]. Another important function of the full-length LHCGR is the capability to form homo- and hetero-dimers [63,64] or homo-oligomers, introducing the possibility of cis- and trans- activation mechanisms [65], as well as constitutive and agonist-dependent self-association mechanism [66]. On the other hand, variants containing exon 6A result in the expression of three diverse types of transcripts (Figure 1) that are present. Three mRNAs can be divided into one terminal transcript and two internal transcripts. The terminal transcript results in a truncated protein (called short-length variant) containing 209 amino acids, whereas internal variants contain a premature stop codon that does not permit their translation (Figure 1). The short LHCGR variant is highly expressed, with values comparable or higher to the full-length LHCGR transcript [67]. The functional role of the LHCGR short variant is not fully understood: it may be capable of hormone binding, considering that this shorter gene segment can encode 8 out of 10 LRRD repeats of the putative extracellular domain present in the full-length receptor, but without showing any signaling activity due to the lack of the transmembrane and intracellular domains [59]. Some authors reported that the full- and the short-length variants of hLHCGR can co-exist in the same cells and that the truncated protein may be secreted acting as a “hormone scavenger” capable of modulating the availability of hCG and LH by binding to them [68]. Conversely, other authors reported that the short variant of hLHCGR, co-expressed with the full-length receptor in transfected 293T cells, shows almost no hormone-binding activity and that its expression is mainly limited to the cytoplasm [69].

The physiological importance of LHCGR in sexual development and reproduction is highlighted by the phenotypic manifestations of activating and inactivating mutations. Indeed, at least 77 damaging variants have been documented for hLHCGR, of which 60 missenses, 3 splicing, 4 small deletions, 4 small insertions, 5 gross deletions, and 1 gross insertion/duplication [70]. In males, a total or partial receptor inactivation can result in Leydig cell hypoplasia (LCH) type I or II, while an increased sensitivity of LHCGR to the ligand or a constitutive and agonist-independent activation can lead to familiar or sporadic male-limited precocious puberty (MPP); furthermore, the somatic gain of function variants has been associated with Leydig cell adenomas. Concerning LCH, it is a rare autosomal recessive disorder of sexual development that is believed to affect 1 in 1,000,000 individuals [71] and is characterized by the total or partial loss of function of LHCGR. Two types of LCH have been defined: LCH type I is the severe form caused by the total inactivation of LHCGR, and it is characterized by complete 46, XY male pseudohermaphroditism with female external genitalia, underdeveloped and retained testes, lack of breast development, amenorrhea, unresponsiveness to LH/hCG, low testosterone and high LH levels in serum, and a lack of secondary male sex characteristics; LCH type II is a milder form in which mutated LHCGR retains some functionality, and it is phenotypically associated with a micropenis, cryptorchidism, and hypospadias. There are a variety of inactivating/partially inactivating variants associated with LCH and all of them can be classified based on the mechanism of inactivation. This means that there are LHCGR variants that show defective biosynthesis (e.g., premature stop codon), while others show defective trafficking towards the endoplasmic reticulum or cell membrane and, finally, defective hormone binding and/or receptor activation. Several mutations affecting the signal peptide (e.g., p.L10P [72,73], p.L16Q, p.L10_Q17dup, p.K12_L15del [74]) or the extracellular and transmembrane domain [75] cause intracellular retention at cytoplasmic or endoplasmic reticulum level, leading to receptor degradation. Other LHCGR variants with mutated extracellular domain usually show reduced or null binding affinity towards both or selectively to one of the two ligands (e.g., p.C131R, p.I152T [75]). An example of selective discrimination among LH and hCG is represented by the clinical case of a patient affected by LCH type II, with a homozygous deletion of exon 10 of *LHCGR* [76]. Mutations affecting the hinge region, transmembrane domain (TMD), and C-terminal intracellular may evolve in an LHCGR variant that is unable to exploit any or sufficient signaling activity (e.g., p.E354K and p.I625K [75]).

The opposite situation is represented by MPP or gonadotropin-independent testotoxicosis, which is a rare autosomal dominant disorder that affects only males. Usually, a mutation causes higher LHCGR basal signaling activity, by a partial structure rearrangement in the TMD that stabilizes and constrains the receptor in the activated form. Boys with this disorder exhibit signs of puberty in early childhood (2–6 years old), with pubic hair growth, acne, enlargement of the penis and testes, and increased height velocity. In these cases, proper anti-steroidogenic treatment is fundamental to prevent virilization and delay the closure of the epiphyseal plates [77]. Approximately 20 activating variants have been identified in the *LHCGR* gene. The most common mutation reported in the literature is p.D578G [78,79]. Interestingly, at this position, other missense variants have been described: p.D578E [80], p.D578Y [81], D578V [82], and p.D578H. Remarkably, p.D578H is somatic in nature and associated with Leydig cell adenomas [78,83]. Another activating mutation causing MPP with high incidence in Brazilian boys is p.A568V [84,85].

It is important to note that some polymorphism may have trivial effects on LHCGR activity and, considering that a large portion of the *LHCGR* gene, including exon 10, is fixed into the intronic region of the TFIIAalpha/beta-like factor (*ALF*) gene that is also associated to male infertility [86], it is possible that the related pathological manifestations are due to the physiological activity of *ALF* gene. In this context, polymorphism has been described, p.S312N and its possible role in men’s infertility [87].

Finally, it is noteworthy that in females, most LHCGR variants have no or minimal effects on reproductive function. However, some inactivating polymorphisms can be associated with oligo/amenorrhea [88], infertility [89], empty follicle syndrome [90], or higher risks of developing polycystic ovarian syndrome [91]; on the other hand, activating variants, such as p.18insLQ, have been reported to be associated with shorter disease-free survival in breast cancer patients [92]. These observations may open up the way to further investigations of a possible link between these mutations and male infertility.

## 4. Mitochondrial Dynamics

Mitochondria are powerhouse organelles involved in many aspects of cell life and metabolism, being the beating heart of every cell. They are dynamic specialized compartments able to rearrange themselves through a continuously coordinated cycle that balances two pivotal events: fusion and fission [93]. The harmony between these two states plays critical roles in maintaining mitochondria function to adapt to cellular metabolic needs: fusion helps mitigate stress by mixing the contents of partially damaged mitochondria and increases their activity, whereas fission is needed to remove suffered and less active mitochondria or to create new ones [94]. The mitochondrial dynamics are enabled by the concerted action of numerous genes and proteins [95]. Although the details of the regulation of mitochondrial fission–fusion dynamics remain to be completely elucidated, its aberrant regulation is observed in various pathophysiological conditions, including cancer, cardiovascular disease, and neurodegeneration, often mirroring deregulated mitochondrial dynamics proteins [96]. Since one of hCG mechanisms of action is to stimulate the reaction chain that starts from cholesterol and brings to steroids and the first steps of steroidogenesis take place in the mitochondria, biological impairments along these reaction steps may negatively influence the effectiveness of the therapy.

### 4.1. Mitochondrial Fission

Mitochondrial fission creates new mitochondria since from a single organelle, two are formed by division. This episode is essential for (i) rapidly dividing and growing cells to populate them with adequate numbers of mitochondria [94], (ii) maintenance of mitochondrial morphology, (iii) a more efficient redistribution of these organelles to the energy-demanding regions, and (iv) mitophagy [97]. Mitochondrial fission is not only an ordinary fixed, unvaried division, but it leads to distinct outcomes depending on the location of the fission event, which in turn is classified into midzone and peripheral fission. Mitochondria that split near the center are more likely to contribute to biogenesis thereafter, whereas the smaller mitochondria that resulted from a split near one of the mitochondrial ends upstream display signs of stress and damage that may lead to degradation [98]. Accordingly, these two peculiar fission patterns derive from differences in membrane polarity, pH, ER-actin contact (reduced in endzone fission), and lysosomal contact (representative of endzone fission) [97]. This is not coincidental, because in fact mitochondrial fission begins with labeling the contact sites of the nucleoid markers of the mitochondrial matrix, the endoplasmic reticulum, and lysosomes [99,100]. Thereafter, outer mitochondrial membrane (OMM) proteins aggregate at a specific site in the OMM. The main actors are mitochondrial fission protein 1 (FIS1), mitochondrial fission factor (MFF), mitochondrial 98 dynamics proteins of 49 kDa (MiD49), and mitochondrial dynamics proteins of 51 kDa (MiD51); focal accumulation of them has the function to attract Dynamin-related protein 1 (DRP1) from the cytoplasm and dock it to the OMM, where oligomeric DRP1 can perform a scissor-like function upon GTP hydrolysis to cut a complete mitochondrion into two separate mitochondria [96,101]. Globally, DRP1 promotes mitochondrial fission stimulating fragmentation of the mitochondrial network, while inhibition of DRP1 expression can lead to highly elongated mitochondria. The mechanism described above is finely tuned by numerous events commonly employed by cells to respond to intracellular and extracellular cues. This strategy can be represented by the following: (1) DRP1 is activated by the focal adhesion kinase (FAK)-mediated phosphorylation at S616; FAK is an environmental biosensor that conveys multiple extracellular signals; (2) similarly, DRP1 is also activated via ERK1/2 pathway mediated by extracellular fibronectin, leading to increased mitochondrial fission, oxygen consumption rate (OCR) and ATP production [102]; (3) the ubiquitous second messenger PKA plays a regulatory role in maintaining mitochondrial activity phosphorylating DRP1 in two distinct sites and leads to the inhibition of mitochondrial fission and reduction in cellular sensitivity to apoptotic stimuli [103,104]; and (4) MFF has been proposed to both activate and oligomerize DRP1 [105]. In addition to the single role of DRP1, FIS1/DRP1 combination causes mitochondrial dysfunction in cells, such as decreased mitochondrial membrane potential, oxidative stress, and bioenergetic failure [98]. For its part, FIS1 can (i) diminish the number and presence of mitochondrial fusion proteins (MFN1, MFN2, and OPA1) by binding to them and fragmenting the mitochondrial network [106,107]; (ii) it can favor and regulate mitochondrial–lysosomal contact that triggers asymmetric mitochondrial division [108,109].

Mitochondrial fission is such an important event that it conditions the cellular state (and further the whole organism) based on its regulation. Indeed, mitochondrial fragmentation in some diseases, including neurodegeneration that is characterized by excessive mitochondrial ROS but low ATP production, is typically thought to result from uncontrolled fission [110]. Moreover, FIS1 overexpression can lead to mitochondrial fragmentation during acute kidney injury [111], promoting mitochondrial division, apoptosis, and pyroptosis of cells [112]. However, it has also been shown that FIS1 is important in regulating mitochondrial autophagy and, in diabetic nephropathy, can play a protective role by participating in mitochondrial quality control through the adaptive mitochondrial autophagy pathway [113], while on the other hand, mitochondrial fission can be reduced by decreasing DRP1 expression to stop disease progression [114]. Finally, DRP1 overexpression is often associated with the presence of tumors, and, in particular, correlates with an aggressive and malignant phenotype as well as appears to be involved in the influence of tumor cell migration/invasiveness [115,116,117,118].

### 4.2. Mitochondrial Fusion

Mitochondrial fusion is the process where one larger mitochondrion is formed from two smaller mitochondria. In this process, there is a fusion of the inner and outer mitochondrial membranes and a mixing of the mitochondrial matrix contents, such as soluble proteins and mtDNA. Mitochondrial fusion is associated with a well-developed mitochondrial network resulting in improved energy efficiency, increased oxidative phosphorylation (OXPHOS) capacity, and maintained ATP production even upon limited nutrients. The mitochondrial network also regulates cell proliferation. Indeed, in the G1 and S phases, mitochondria are elongated, providing enough ATP for cell replication. Mitochondrial fusion also allows the exchange of mitochondrial contents between adjacent mitochondria, thus facilitating the maintenance of mtDNA levels and functional complementation in mitochondrial deficiencies [119]. In mammals, mitochondrial fusion is a two-step process: fusion of the OMM is mediated by two mitofusins (MFN1 and MFN2), whereas the inner mitochondrial membrane (IMM) is mediated by optic atrophy 1 (OPA1). Both are GTPase dynamin-like proteins. The function of both mitofusins is essential for embryonic development due to the fact that mice deficient in either MFN1 or MFN2 die in midgestation, but MFN1 and MFN2 are not completely functionally redundant since deficiency of MFN1 or MFN2 shows different forms of fragmented mitochondria in embryonic fibroblasts [120]. These findings suggest that MFN1 and MFN2 may have distinct functions outside of their principal roles in mitochondrial fusion. MFN2 is required for glucose homeostasis [121], steroidogenesis [95], cerebellar development and function [122], autophagy [123], and coenzyme Q homeostasis [124]. Several recent findings correlate mutations/alterations in MFN1-2 with metabolically related disorders, neurodegenerative diseases, and cardiovascular diseases [119]. The expression of mitofusins is regulated by the transcription factors involved in mitochondrial biogenesis and oxidative phosphorylation. Both PGC-1α and PGC-1β stimulate MNF2 expression by targeting the *MFN2* promoter under a variety of physiological conditions (metabolic demand, exercise, and exposure to cold). In particular, PGC-1α or PGC-1β directly binds to the nuclear receptor Estrogen-Related Receptor α (ERRα) and co-activates the transcription of MFN2 [125,126]. Concerning OPA1, it contains a mitochondrial targeting sequence that is removed within the mitochondrion, a transmembrane domain, which allows anchoring to IMM, a coiled-coil domain, the GTPase domain, and the GTPase effector domain at the C-terminal. The eight different isoforms are ubiquitously expressed but are present in different amounts depending on the tissue type [127]. OPA1 present as long form (L-OPA1) is anchored to IMM when it is processed by proteases and is soluble in intermembrane spaces (S-OPA1). The main proteases implicated in the L-OPA1 cleavage are OMA1 and YME1L. Under normal conditions, YME1L is constitutively active and generates basal levels of S-OPA1, producing a steady-state balance of long and short OPA1 isoforms. Under stress conditions and mitochondrial dysfunction, stress-sensitive OMA1 actively determines an accumulation of fusion-inactive S-OPA1, causing mitochondrial network collapse to a fragmented population of mitochondria [128]. For IMM fusion is critical a role of cardiolipin, a mitochondrial lipid. The interaction between L-OPA1 and cardiolipin on either side of the membrane tethers the two IMM, which fuse following OPA1-dependent GTP hydrolysis [129]. S-OPA1 alone is not sufficient to promote fusion but accelerates the L-OPA1-dependent fusion activity and promotes liposome binding, suggesting that S-OPA1 may support a bridge between L-OPA1 and cardiolipin on opposite membranes [129]. OPA1 is also involved in determining the shape of mitochondrial cristae being a key player for the MICOS complex and the F1Fo-ATP synthase in cristae biogenesis and maintenance [130]. Heterozygous mutation in gene *OPA1* results in autosomal dominant optic atrophy, an inherited neuropathy characterized by loss of retinal ganglion cells and optic nerve fibers, resulting in progressive loss of visual acuity. In addition, heterozygous *OPA1* mutations are also associated with other symptoms including deafness, ataxia, axonal sensory-motor polyneuropathy, and mitochondrial myopathy [119].

## 5. Implications of Mitochondrial Dynamics in Male Infertility

A growing number of research articles describe the relationship between mitochondrial dynamics and fertility alterations. Indeed, recent discoveries revealed that several reproductive diseases, including polycystic ovary syndrome for women and asthenozoospermia for men, are associated with alterations in mitochondrial dynamics with an interest in the reduced levels of MFN2 [131]. Due to the widely described central role of mitochondria in sex steroid hormone synthesis and in the proper maturation of germ cells in males, in this chapter we report the studies describing the causal link between steroidogenesis or spermatogenesis and alterations of mitochondria structure, corroborating the importance to explore this field for the identification of new potential targets of impaired fertility.

### 5.1. Mitochondrial Dynamics Regulate Steroidogenesis in Male Cells

In males, steroid synthesis takes place in a sub-type of testicular cells, i.e., Leydig cells, starting from cholesterol as the first substrate. There are different sources of cholesterol, including de novo synthesis, cholesteryl-esters storage, and lipoproteins [132]. Concerning the latter, it is worthwhile to note that a class of lipoproteins, i.e., low-density lipoprotein (LDL), are recognized by Leydig cells thanks to the surface expression of the relative receptor (LDLR), which has been reported to be expressed at high levels by this testicular cell type, in addition to endothelial and peritubular cells that also compose the testis (source: The Human Protein Atlas website). Indeed, it has been shown that serum testosterone levels are reduced in an *LDLR*-knockout (k.o.) mouse model in comparison to wild-type (wt) mice, together with a diminished number and size of Leydig cells [133]. Once cholesterol is inside Leydig cells, it is stored in lipid droplets and, then, it must enter mitochondria where the first enzymatic reaction of steroidogenesis takes place (Figure 2). Already in this phase, it has been reported that MFN2 favors lipid droplet-mitochondria contact, supporting cholesterol transfer to the organelle [134]. In this process, the limiting steps comprise different protein-protein interactions that support the translocation of cholesterol into the mitochondria and its conversion into pregnenolone. Within this context, a correct conformation in mitochondria of the double membrane-bound and of the cristae is necessary for steroidogenesis since proteins that are involved in cholesterol transport and in pregnenolone generation are anchored to them. Indeed, the protein complex called transduceosome brings cholesterol from the cytosol to the OMM, where it reaches the enzyme CYP11A1 through the outer/inner membrane contact sites. StAR is a protein of the transduceosome that favors the cholesterol transport into the mitochondria and is critical for hormone-induced steroid production that, in turn, increases its expression and phosphorylation [135]. In a study, it has been demonstrated that the enhancement of StAR expression is linked to an increase in steroidogenesis and that its k.o. delays sperm maturation and induces germ cell apoptosis in the testes of mice [136]. Galano et al. recently demonstrated that the depletion of StAR expression alters mitochondrial morphology and cristae structure in a mouse cell line of Leydig cells [137]. In particular, they showed that the presence of StAR in the OMM is strictly connected with a proper mitochondrial structure and function and that dysfunctions of this organelle lead to defective StAR presence at the OMM. In another study, it has been demonstrated that the phosphatase SHP2, a key protein for steroidogenesis [138] whose expression is directly proportional to StAR’s expression, also regulates mitochondria fusion and that MFN2 knockdown is sufficient to impair steroid synthesis [95]. In line with this, the up-regulation of MFN2 in MA-10 cells has been shown to be essential for maximal steroid production, and ERK-mediated phosphorylation of StAR is required to retain the protein in mitochondria [95,139].

In addition to StAR, also CYP11A1 function is strictly dependent on the correct assembly of mitochondria since this enzyme is present on the IMM, where it cleaves a bound responsible for the transformation of the soluble pregnenolone, which then activates steroidogenic enzymes to produce testosterone [140]. Shan et al. tested the effect on progesterone biosynthesis of the endocrine disruptor polybrominated diphenyl ethers (PBDE) in BeWo cells [141]. They showed that this compound decreases MFN2 and enhances FIS1 and DRP1 expression, thus modifying mitochondria morphology by increasing their fragmentation. Despite the authors showing that the effect of this alteration is connected with decreased progesterone production in a dose-dependent manner, their results indicate that CYP11A1 activity is not involved in this process [141]. Conversely, in a previous study it has been shown that the same class of chemical compounds affect progesterone biosynthesis in a mouse Leydig tumor cell line (mLTC-1) by decreasing mRNA and protein expression of CYP11A1 [142], thus leaving the question of how diverse cell types could respond differently to the same compound open.

Finally, a very recent paper demonstrated that, in mice, the blockade of the androgen receptor (AR) by flutamide, an anti-androgen compound, increases the expression of StAR and CYP11A1, together with an alteration of the morphology of Leydig cell, which appeared to be larger than in control cells, and of their mitochondria, which are more elongated and swollen [143]. In addition, since flutamide elevates the levels of testosterone and estradiol, the authors suggest that this compound leads to the inhibition of the negative feedback of androgens on the HPG axis, enhancing Leydig cell steroidogenic activity [143].

Altogether, these studies highlight that mitochondria shape, in particular their elongated form, represents a crucial regulation point for the transport of intermediate products inside and outside mitochondria, being essential for the correct cholesterol import into these organelles and for steroid biosynthesis.

### 5.2. Mitochondrial Dynamics in Spermatogenesis

In different articles, it has been demonstrated that, during sperm formation, mitochondrial fusion plays a key role with MFNs as important players in this scenario by supporting the ATP oxidative phosphorylation to obtain the energy necessary for spermatogenesis. In particular, MFN2 has been shown to have a role during spermatogenesis in different animal models. Indeed, recently, through the use of conditional k.o. mouse model lacking MFN2, it has been demonstrated that seminiferous tubules present an empty structure due to disrupted spermatogenesis and increased apoptosis. In addition, the mitochondria respiratory chain (ETC) is also affected since the expression of the subunit of ETC complexes is lower in k.o. mice in comparison to control animals, supporting an interference of cellular respiration [144]. In another study that used the same animal model lacking MFN2, it has also been demonstrated that the k.o. led to male sterility characterized by mitochondria defects and structurally abnormal MAMs, which are the contact points between mitochondria and ER membranes that take place during the fusion process [144]. By analyzing the roles of MFN2 and MFN1, the same authors showed that MFN1 is also essential for spermatogenesis but has a distinct role in comparison to MFN2. Indeed, they suggested that the interaction between MFN2 and MFN1 regulates the distribution of mitochondria in the testes, thereby contributing to spermatogenesis and male germ cell development, although the ablation of MFN1 alone is not able to disrupt the MAM and ER structures in male germ cells [144]. Additional information about the role of MFN1 in this process has been given by the demonstration that this protein interacts with GASZ, a germ cell-specific protein with a mitochondrial targeting sequence that leads to its dimerization at the OMM and to its collaboration with MFN1 to promote the fusion process; GASZ loss has been shown to cause male infertility [145].

Interestingly, in another study, it has been shown that MFN1 and MFN2 affect spermatogenesis in distinct manners: indeed, in neonatal pro-spermatogonia, only MFN1 regulates mitochondrial fusion, whereas later in spermatogenesis, the loss of MFN2 impairs such an event. These data support the important role of both MFNs in mediating mitochondrial fusion and ER homeostasis, sustaining male fertility in a non-redundant manner [146]. A biochemical approach revealed that spermatogenesis arrest due to the depletion of MFN1 and MFN2 in mice is related to metabolic alterations. Indeed, during sperm maturation, the energy metabolism is shifted from glycolysis to oxidative phosphorylation, together with an increase in mitochondrial content and elongation. In line with this, during the final step of spermatogonia differentiation, high levels of mitochondrial fusion are detected in addition to active oxidative phosphorylation, supported by the aberrant cristae ultrastructure and the reduced expression of complex I and IV subunits in mitofusin-deficient germ cells [147].

In contrast to the data showing that disturbed mitochondrial fusion by deletion of *MFN1/MFN2* specifically blocks spermatogonial differentiation, another study demonstrated that the double knockout of the two *MFNs* does not affect sperm development and fertility in post-meiotic haploid germ cells. On the other side, authors suggested that mitochondrial fusion is dispensable for sperm development but only during late spermatogenesis, highlighting the importance of studying mitochondrial regulation along with various developmental stages of spermatogenesis [148].

Conversely, fewer studies demonstrated that mitochondrial fission could exert a role during the differentiation of male germ cells. Indeed, it has been demonstrated that, in addition to mitochondrial fusion, the fission process is also important for regulating the maintenance of early germ cells in *Drosophila* larval testis [149]. The authors showed that inhibition of DRP1, the key regulator of fragmented mitochondria, resulted in the loss of germline stem cells and spermatogonia due to the accumulation of reactive oxygen species (ROS) and the activation of the epidermal growth factor receptor (EGFR) pathway in adjacent somatic cells, demonstrating that the fission event is necessary for the maintenance of spermatogonia in the larval testis. This study concludes that, in *Drosophila*, both mitochondrial fusion and fission are important for the preservation of germline stem cells by activating different downstream cellular response pathways, opening the way for further investigations of this balance even in mammals [149]. In addition to DRP1, also the protein FIS1, which is involved in the fragmentation of the mitochondrial network and in the recruitment of DRP1 on the organelle surface, has a role in the regulation of spermatid maturation [150]. Indeed, by using conditional FIS1 mice, the authors showed that the ablation of this protein brought early spermatid development arrest, highlighting that FIS1 is required for spermatogenesis and mitophagy. Finally, a study performed on MFF mutant mice showed that this protein is necessary for the correct organization of the mitochondrial sheath in mouse sperm [151].

Altogether, these results accentuate the importance of the maintenance of a proper mitochondrial shape during the different stages of spermatogenesis, including both fusion and fission events (Table 3). However, the animal models used to perform these studies are different, from flies to mice, rendering necessary a translation of data on human cells.

### 5.3. The Effect of Hormonal Therapies May Correlate with Mitochondrial Assembly

As reported in the previous chapters, a strict link between steroidogenesis and mitochondria dynamics exists, thus rendering plausible the influence of a proper mitochondrion assembly on the effectiveness of hormonal therapy, specifically hCG. Indeed, the binding of hCG to LHCGR stimulates intracellular pathways that bring about steroid synthesis; hence, an intact structure of mitochondria, within which the first and rate-limiting step of TT production takes place (Figure 2), represents an important piece of the puzzle.

In support of this consideration, it has been demonstrated that MA-10 Leydig cells present more punctuated mitochondria with narrow cristae in comparison to the same cells treated with hCG, which show more fused, large, and tubular mitochondrial structures [95]. In addition, the same authors also showed that hCG increases MFN2 expression and its knockdown is sufficient to impair steroid biosynthesis [95], supporting a key role of mitochondrial fusion during steroidogenesis and hormonal stimulation of this process. Another study demonstrated this connection in in vivo and ex vivo animal models showing that the hCG analog (i.e., LH) increased mitochondrial mass, mtDNA, oxygen consumption, cAMP, CYP11A1, and StAR expression, together with a decreased expression of the key fission player, i.e., DRP1 [140]. Interestingly, Leydig cells derived from hypogonadotropic-hypogonadal animals and treated with LH showed low cAMP production and mtDNA content, as well as a reduced expression of mitochondria fusion markers (i.e., MFN1/2 and OPA1) [140]. Interestingly, it has been shown that pathological mitochondria fission may influence the translocation of PKA, which is involved in the signal cascade of LHCGR as described above, inside mitochondria [147] with consequences on the signaling pathway, alterations of energy metabolism, cell proliferation, and induction of vasculogenesis.

## 6. Conclusions

In order to define effective therapies for patients, it is crucial to merge the clinical investigation with the search for the biological cause, even in a more personalized way; only in this way can the identification of the best therapeutic approach be successful. The early prevention of fertility impairment is a challenge that would derive benefits from the identification of the proper therapy for each patient. In this context, hCG (or other hormonal therapies) is widely used as adjuvant therapy after orchiopexy or for the treatment of infertility in adulthood; however, it would be important to define whether the therapy would not be successful due to alterations in the relative receptors (including mutations) or variations in mitochondria morphology/dynamics that could impair the signal cascade. At the present, as described above, only a few papers show that mitochondria fusion supports steroidogenesis and spermatogenesis. However, there is still a lack of knowledge on the link between mitochondria arrangement and decreased TT production, spermatozoa function, cell proliferation, and vascularization in patients with impaired fertility potential. In addition, some controversial evidence emerged about the balance of fission/fusion processes in spermatogenesis. Hence, an extensive study of these aspects would help to improve the effectiveness of hormonal therapy, suggesting the possibility of combining this treatment with drugs that sustain a proper mitochondrial arrangement. In this context, since fusion emerges to play a potentially crucial role in male fertility and hormonal therapy effectiveness, this process may represent a prospective candidate for combined therapy. Some drugs that increase mRNA expression or activation of MFN1/MFN2 have been described in in vivo models, thus opening the way to their application for other diseases [153,154,155]. Nevertheless, it is important to consider that this therapy should be tissue-specific to avoid a general alteration in mitochondrial homeostasis. In conclusion, a deep investigation of mitochondria shape and functionality could be necessary for identifying the severity of impaired fertility potential and for adopting the best therapeutic approach based on patient parameters.

## Figures and Tables

**Figure 1 biology-12-00547-f001:**
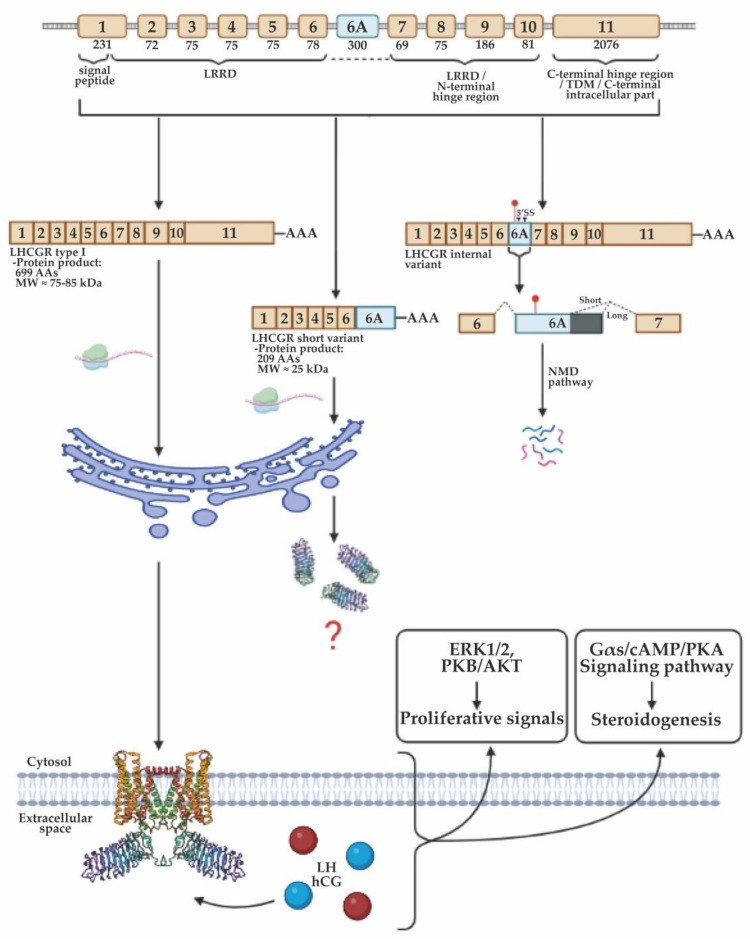
Transcription and translation of LHCGR variants. From top to bottom, we graphically report the complete *LHCGR* gene assembly, the putative transcriptional model, and the receptor structure/signaling pathways activated by LHCGR. In the schematic representation of the internal variant, the red spot indicates the translational stop codon, while 5′ SS indicates the two 5′ splicing sites that give rise to the internal-short and internal-long variants, both degraded by nonsense-mediated mRNA decay (NMD) pathway. The 3D monomeric LHCGR structural models, here reported as dimer on the cellular surface, have been uploaded from Protein Data Bank (protein #7FIJ).

**Figure 2 biology-12-00547-f002:**
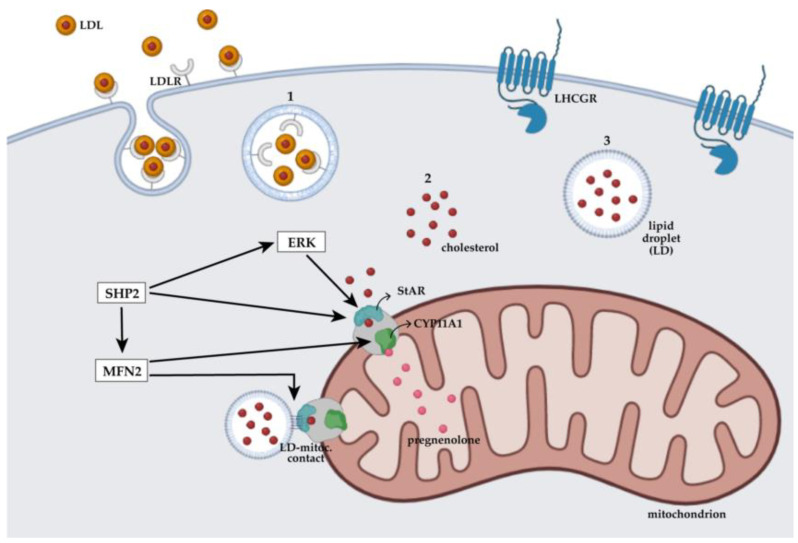
Steroidogenesis regulation in Leydig cells. The first critical step for the synthesis of steroids is the accumulation of cholesterol inside the cell. Here we represent a Leydig cell, expressing the specific receptor LHCGR. This cell type can obtain cholesterol in three ways: (1) through its transport via low-density-lipoprotein (LDL, in orange), captured thanks to its respective receptor LDLR and subsequently internalized in vesicles; (2) de novo synthesis in the cytosol; (3) accumulation in lipid droplets (LD). At this point, the cholesterol must be internalized in mitochondria for the first step of steroidogenesis to take place, and this translocation occurs thanks to StAR protein (in blue), which in turn is part of the so-called transduceosome (in gray). This complex is present on the mitochondrial membrane, highlighting the importance of their arrangement, which is finely regulated by MFN2, a known protagonist of mitochondrial fusion. In this context, MFN2 allows contacts between the lipid droplets and the mitochondrial membrane to further ensure the transport of cholesterol inside the organelle, where it reaches CYP11A1 (another component of the transduceosome, shown here in green) which transforms it into pregnenolone. This whole pathway is regulated by additional agents: (i) SHP2, a phosphorylase that acts on MFN2, ERK, and StAR, and (ii) ERK ensures the correct position of StAR on the mitochondrial membrane.

**Table 1 biology-12-00547-t001:** Serum hCG levels during gestation.

Period of Life		hGC Serum Levels (UI/L)	References
Gestation	0–1 week	0–50	[25]
	1–2 weeks	40–300	
	3–4 weeks	500–600	
	1–2 months	5000–200,000	
	2–3 months	10,000–100,000	
	second trimester	3000–50,000	
	third trimester	1000–50,000	

**Table 2 biology-12-00547-t002:** Serum LH levels in men, from fetal life to adulthood.

Period of Life		LH Serum Levels (UI/L)	References
Gestation	0–1 weeks	no detectable	[51]
	1–2 weeks		
	3–4 weeks		
	1–2 months		
	2–3 months	20	[52]
	second trimester	26.1	
	third trimester	<second trimester	
Birth			
Infancy “mini-puberty”	after 2 days of birth	0.21	[53]
	after 7 days of birth	3.94	
	after 10 days of birth	4.81	
	after 20 days of birth	2.67	
Childhood	1–10 years of life	no detectable	[49]
Puberty	Tanner stage I	0.02–0.42	[54]
	Tanner stage II	0.26–4.84	
	Tanner stage III	0.64–3.74	
	Tanner stage IV	0.55–7.15	
	Tanner stage V	1.7–8.6	
Adulthood	18–30 years of life	1.8–8.6	[50]
	50 years of life	2.1–10.4	
	70 years of life	2.22–11.2	

**Table 3 biology-12-00547-t003:** Effect of proteins directly or indirectly involved in the regulation of mitochondrial dynamics on spermatogenesis or steroidogenesis.

Protein	Biological Effect	Physiological Effect	Model/Cell Type	References
DRP1	Maintenance of spermatogonia	spermatogenesis	Spermatogonial cells (*Drosophila*)	[149]
FIS1	Regulation of spermatid maturation	spermatogenesis	Conditional mouse model	[150]
MFF	Determination of mitochondrial sheath	spermatogenesis	Mutant mouse model	[151]
MNF1	Regulation of mitochondrial fusion in neonatal pro-spermatogonia	spermatogenesis	K.O. mouse	[146]
MNF1-MNF2 interaction	Support of mitochondrial distribution in the testes	spermatogenesis	Postnatal male germ cells (mouse)	[144]
MNF2	Support of lipid droplet-mitochondria contact	steroidogenesis	BeWo cells (human)	[134]
	Increment of progesterone production and CYP11A1 expression	steroidogenesis	BeWo (human), mLTC-1 (mouse) cells	[141,142]
	Support of spermatogenesis and filling of seminiferous tubes	spermatogenesis	Conditional K.O. mouse model	[152]
	Regulation of mitochondrial fusion late steps of spermatogenesis	spermatogenesis	K.O. mouse	[146]
SHP2	Support of mitochondrial fusion	steroidogenesis	MA-10 cells (mouse)	[138]
	Regulation of ERK1/2 localization in the mitochondria	steroidogenesis	MA-10 cells (mouse)	[95]
StAR	Conferment of mitochondria and cristae proper conformation	steroidogenesis	MA-10 cells (mouse)	[137]

## Data Availability

No new data were created for this study.

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
