# Peer review of "Mitochondrial Dynamics as Potential Modulators of Hormonal Therapy Effectiveness in Males"

_biology, 2023, doi:10.3390/biology12040547_

Round 1
Reviewer 1 Report
Andrea Errico et al.'s review summarizes recent data on male fertility diseases and their relationship with mitochondrial quality control processes, specifically mitochondrial fusion, and fission. Overall, the authors did a good job summarizing the different mechanisms involved in andrological diseases. However, some major and minor issues must be addressed, particularly in the mitochondrial dynamics chapters.
Major
The review's title about mitochondrial dynamics as a potential modulator of hormonal therapy effectiveness in males is not supported by the cited literature. The last chapters, as a review of the roles of MFN1 and MFN2 and their role in male fertility, are acceptable. Still, the authors fail to show literature on MFN1 or MFN2 used as a therapeutic or potential therapeutic approach. In the last five years, there have been some attempts to create MFN1 agonist drugs and peptides to boost mitochondrial fusion. Why do the authors not mention these drugs? Some drugs also inhibit mitochondrial fission, and they were not mentioned.
The manuscript uses too many reviews as a reference, and since this is a literature review, it should be able to provide and cite the primary data. Without this amendment, it makes very difficult to evaluate the text.
The review is mainly focused on the role of MFN1 and MFN2, and there is some mention of the DRP1 role. Why did the authors not consider the part of other essential components of mitochondrial fusion and fission, such as OPA1, MFF, MID49/MID51, and ER-Mitochondria contacts, among others?
Also, the figures are not used well throughout the text and fail to help the reader because they are used to explain straightforward concepts. I would suggest the authors improve the figures in a way they to be harmonized with the text. Also, I would recommend creating a table to summarize all the findings related to the proteins involved in mitochondrial dynamics and their effect on male fertility.
Minor
Overall, the review is not written linearly, and sometimes the authors go very deep into details that are not relevant to the topic of the study. An example is the details of MFN1 and MFN2 structures that are not covered in the following chapters.
Due to the roles of MFN2 in the ER-Mitochondria contacts, I think it would be helpful to consider a chapter that could cover the relevance of ER-mitochondrial communication for the dynamics and their potential role in the andrological diseases.
The authors emphasize the role of FIS1 when it has been shown that it is not very specific for the mitochondrial division in mammals since it is also involved in peroxisome fission. Why do the authors not mention MFF, MID49/MID51?
Is there any data on MFN2, OPA1, DRP1 patients, and male infertility?
Author Response
Major
The review's title about mitochondrial dynamics as a potential modulator of hormonal therapy effectiveness in males is not supported by the cited literature. The last chapters, as a review of the roles of MFN1 and MFN2 and their role in male fertility, are acceptable. Still, the authors fail to show literature on MFN1 or MFN2 used as a therapeutic or potential therapeutic approach. In the last five years, there have been some attempts to create MFN1 agonist drugs and peptides to boost mitochondrial fusion. Why do the authors not mention these drugs? Some drugs also inhibit mitochondrial fission, and they were not mentioned.
Response: The aim of our review was to emphasize the concept that patients may respond differently to hormonal therapy due to alterations of mitochondria arrangement in their testicles, thus influencing the effectiveness of the therapy. We stressed the concept that mitochondria dynamics may influence steroidogenesis and spermatogenesis, hence an analysis of mitochondria structure in the testis should be taken into account in order to adopt the best therapeutic approach based on each patient’s parameters. Nevertheless, we thank the reviewer for the important suggestion: we implemented the conclusion of our review by also suggesting to combine hormonal therapy with drugs that support mitochondrial fusion in case of altered fission. However, we have to highlight our concerns about the necessity to stimulate mitochondrial fusion with tissue specific drugs, in order to avoid a general alteration of mitochondrial homeostasis.
The manuscript uses too many reviews as a reference, and since this is a literature review, it should be able to provide and cite the primary data. Without this amendment, it makes very difficult to evaluate the text.
Response: We thank the reviewer for the suggestion, we have updated many of the references, using papers reporting the primary data. However, some reviews have been maintained since we also took into account or reported the point of view that emerged by the authors.
The review is mainly focused on the role of MFN1 and MFN2, and there is some mention of the DRP1 role. Why did the authors not consider the part of other essential components of mitochondrial fusion and fission, such as OPA1, MFF, MID49/MID51, and ER-Mitochondria contacts, among others?
Response: In chapter 4, we generally described the key players of mitochondrial fusion and fission processes, including the proteins reported by the reviewer. However, there are not yet evidence in the literature about the involvement of all of them in the regulation of steroidogenesis and spermatogenesis. Thus, in chapter 5 we described mainly MFN2 and MFN1, since the published papers about the connection of mitochondrial dynamics and steroidogenesis/spermatogenesis are focused on these two proteins. It is reasonable that future studies will show that also OPA1, MFF, Mid49/Mid51 and ER-mitochondria contacts are connected with steroidogenesis and spermatogenesis. However up to now, to our knowledge, no papers show this link through experimental data, exception for MFF and spermatogenesis, which reference and paper description has been reported in line 806 of the revised version of the manuscript.
Also, the figures are not used well throughout the text and fail to help the reader because they are used to explain straightforward concepts. I would suggest the authors improve the figures in a way they to be harmonized with the text. Also, I would recommend creating a table to summarize all the findings related to the proteins involved in mitochondrial dynamics and their effect on male fertility.
Response: We decided to design the two figures based on the more complicated aspects discerned in the review. Indeed, Figure 1 reports the discoveries about the transcript variants and isoforms of the gene LHCGR by taking into account different papers that investigated diverse aspects of this gene/protein. In the literature the variants of this receptor are still under investigation, thus we consider that the recent discoveries about LHCGR could be summarized in a schematic representation in order to help the reader to have a wider vision of this receptor. In the same way, we designed Figure 2 with the aim to schematically represent the “core” of the review: indeed, in this figure we considered all the discoveries, which are also described in the text, that highlight a connection between mitochondrial dynamics proteins and steroidogenesis (including the direct or indirect connection of SHP2, ERK, MFN and the proteins CYP11A1 and StAR that have a crucial role in steroidogenesis). To our opinion, the concepts reported in the figure are not such simple and, as commented by another reviewer, we consider that the figures could help in the comprehension of the text. Anyhow, as requested by the reviewer, we agree with the fact that a table that summarize the connection between mitochondrial dynamics proteins and steroidogenesis/spermatogenesis may help the reader to have a clear vision of the link between them. Hence, we prepared Table 3, which reports the direct or indirect connection of proteins involved in mitochondrial dynamics and the steroidogenic/spermatogenic pathways.
Minor
Overall, the review is not written linearly, and sometimes the authors go very deep into details that are not relevant to the topic of the study. An example is the details of MFN1 and MFN2 structures that are not covered in the following chapters.
Response: We thank the reviewer for the suggestion to improve our manuscript and to cut information that are not essential for the comprehension of the entire paper. We have brought these modifications throughout the manuscript, especially in the chapter of mitochondrial dynamics.
Due to the roles of MFN2 in the ER-Mitochondria contacts, I think it would be helpful to consider a chapter that could cover the relevance of ER-mitochondrial communication for the dynamics and their potential role in the andrological diseases.
Response: the point raised by the reviewer is very important in the field of mitochondrial dynamics’ regulation and mechanism of action. However, no relevant papers are reported regarding this part and andrological diseases, hence we decided not to speculate on this aspect in absence of clear experimental evidence. Thus, in order to maintain the manuscript as linear as possible, we avoided to add this part in the manuscript, still considering this point as crucial and as a future investigation aspect.
The authors emphasize the role of FIS1 when it has been shown that it is not very specific for the mitochondrial division in mammals since it is also involved in peroxisome fission. Why do the authors not mention MFF, MID49/MID51?
Response: We agree with the reviewer about the double role of FIS1. However, since the evidence about FIS1 role in andrological disease has been described (as we reported in chapter 5.2), we considered important to describe more in depth this protein in the chapter of mitochondrial dynamics, despite other fission factors (like MFF, MI49 and MID51). As a further check, we controlled again whether some new papers have been published within these last weeks on MID49 or MID51 and andrology/andrological disease, but no papers are present. Whereas, about MFF we added a paper in line 806 of the revised version of the manuscript.
Is there any data on MFN2, OPA1, DRP1 patients, and male infertility?
Response: We have not found any papers about MFN2 and male infertile patients, as well as for OPA1 and DRP1. This molecular aspect is of crucial interest in our opinion, since we think that it would be very important to deep these aspects also on patients, not only on animal models or cell lines. Indeed, the few papers present in literature about this point have been studied on women with fertility issues, not in men, thus further highlighting the low interest about the “male infertility sphere” that is emerging to cover about half of infertile couples, needing more investigations at molecular levels.
Reviewer 2 Report
The manuscript is an interesting review of hormonal functions and mitochondrial function in the male reproductive system and function. Additionally, there was an interesting review on treatments to some of the subfertility issues seen in men.
It would have been helpful to have line numbers.
Simple summary
The simple summary shows the importance of the paper well, however, it does not give an insight on what diseases will be discussed. Add a better description of the general diseases that will be discussed.
It seems like you will only talk about pediatric cases, edit this.
Abstract
Change the term “treat urological disclosures”
Add literature evidence of cryptorchidism being hereditary.
Rewrite the sentence: in which intervene before the occurrence of permanent ischemic damage; retard
Review and rewrite, such as these sentences: Notably, risk factors that could alter the fertility
despite different TT formulations for replacement therapy show significant
levels drop until the birth and remain
Better clarify the importance of this data (Serum levels of hCG change during the male’s life) to your manuscript
The figures help to elucidate the concepts.
When you mention subfertility or infertility are there numbers/percentages that you can add from the literature?
Author Response
It would have been helpful to have line numbers.
Response: We added line numbers to the revised version of the review.
Simple summary
The simple summary shows the importance of the paper well, however, it does not give an insight on what diseases will be discussed. Add a better description of the general diseases that will be discussed. It seems like you will only talk about pediatric cases, edit this.
Response: In the revised version of the simple summary, we clarified that andrological diseases affecting both pediatric and adult patients are discussed in the review.
Abstract
Change the term “treat urological disclosures”
Response: Amended.
Add literature evidence of cryptorchidism being hereditary.
Response: We thank the reviewer for the suggestion that implements the description of cryptorchidism origin. We added this part starting from line 54 of the revised version of the manuscript.
Rewrite the sentence: in which intervene before the occurrence of permanent ischemic damage; retard
Response: Amended.
Review and rewrite, such as these sentences: Notably, risk factors that could alter the fertility despite different TT formulations for replacement therapy show significant levels drop until the birth and remain
Response: Amended.
Better clarify the importance of this data (Serum levels of hCG change during the male’s life) to your manuscript
Response: we thank the reviewer for the suggestion. We have implemented the description of this data, also including the importance of hCG detection in adult age in case of neoplasia. Please, see the revised text from line 189.
The figures help to elucidate the concepts.
Response: We thank the reviewer for appreciating the figures.
When you mention subfertility or infertility are there numbers/percentages that you can add from the literature?
Response: we really thank the reviewer for the suggestion to implement the review with data about worldwide infertility and, more specifically, male infertility. We added this information in line 90 of the revised version of the review.
Reviewer 3 Report
In this review article, Errico and colleagues summarized the current knowledge of andrological diseases, with an emphasis on mitochondrial dynamics in male infertility. This article is very well written and of general interest to reproductive biologists and andrologists. I support publication of this paper in Biology pending typo corrections.
1. “with a frequency that is about 5% of full-term newborns and 30% of
1.premature newborns”, please provide references for this statement.
2. “and may negatively influence the fertile potential”, change “fertile potential” to “fertility potential”.
3. Change “Notably, that risk factors that” to “Notably, the risk factors that”.
4. “Infertility is a worldwide clinical and social problem that affects an increasing number of people each year worldwide”, there are two “worldwide” in this sentence, remove one of them.
5. Consider removing “indicating that there are several conditions and risk factors”. It reads redundant.
6. Change “germinal cells” to “germ cells”.
7. Change “Indeed, hCG adjuvant therapy has been shown to improve testicular vascularization, volume, and morphology, resulting safe and useful” to “Indeed, hCG adjuvant therapy has been shown to safely improve testicular vascularization, volume, and morphology”.
8. Italic is needed when referring to genes. Please correct the gene symbols throughout the text.
9. Human Protein Atlas provides single-cell expression of each gene. It would be helpful if you emphasize that the expression of LDLR is restricted to supporting cells in testis.
10. “At the present, some papers show that mitochondria fusion supports steroidogenesis and spermatogenesis”. References are needed here.
Author Response
In this review article, Errico and colleagues summarized the current knowledge of andrological diseases, with an emphasis on mitochondrial dynamics in male infertility. This article is very well written and of general interest to reproductive biologists and andrologists. I support publication of this paper in Biology pending typo corrections.
- “with a frequency that is about 5% of full-term newborns and 30% of premature newborns”, please provide references for this statement.
Response: we thank the reviewer for the observation. We have added the reference to the text.
- “and may negatively influence the fertile potential”, change “fertile potential” to “fertility potential”.
Response: Amended
- Change “Notably, that risk factors that” to “Notably, the risk factors that”.
Response: Amended
- “Infertility is a worldwide clinical and social problem that affects an increasing number of people each year worldwide”, there are two “worldwide” in this sentence, remove one of them.
Response: Amended
- Consider removing “indicating that there are several conditions and risk factors”. It reads redundant.
Response: Amended
- Change “germinal cells” to “germ cells”.
Response: Amended
- Change “Indeed, hCG adjuvant therapy has been shown to improve testicular vascularization, volume, and morphology, resulting safe and useful” to “Indeed, hCG adjuvant therapy has been shown to safely improve testicular vascularization, volume, and morphology”.
Response: Amended
- Italic is needed when referring to genes. Please correct the gene symbols throughout the text.
Response: we thank the reviewer for the correct observation. We have adjusted the style of the gene with italic.
- Human Protein Atlas provides single-cell expression of each gene. It would be helpful if you emphasize that the expression of LDLRis restricted to supporting cells in testis.
Response: We are grateful to the reviewer for her/his suggestion. Indeed, despite this receptor is expressed in different cell types due to its importance, we added additional details about the testicular cell type that express it, based on the single-cell expression database of Human Protein Atlas. Please, see line 671 of the revised version of the manuscript.
- “At the present, some papers show that mitochondria fusion supports steroidogenesis and spermatogenesis”. References are needed here.
Response: We have checked in which part of the manuscript is reported the part cited by the reviewer. We found it only in the conclusion, which generally does not require references since it represents a final part summarizing the key concepts of the manuscript. However, to make it clearer, we changed this part as follows: “At the present, as described above, only few papers show that mitochondria fusion supports steroidogenesis and spermatogenesis…” – line 853 of the manuscript
Round 2
Reviewer 1 Report
The authors have addressed most of my concerns.
Congratulations to the authors, they did a very nice job and the chapter on mitochondrial dynamics has been substantially improved!